# Optical Properties of a Tapered Optical Fiber Coated with Alkanes Doped with Fe_3_O_4_ Nanoparticles

**DOI:** 10.3390/s22207801

**Published:** 2022-10-14

**Authors:** Karol A. Stasiewicz, Iwona Jakubowska, Joanna E. Moś, Paweł Marć, Jan Paczesny, Rafał Zbonikowski, Leszek R. Jaroszewicz

**Affiliations:** 1Faculty of Advanced Technologies and Chemistry, Military University of Technology, 2 Kaliskiego St., 00-908 Warsaw, Poland; 2Institute of Physical Chemistry, Polish Academy of Sciences, Kasprzaka 44/52 St., 01-224 Warsaw, Poland

**Keywords:** tapered optical fiber, magnetic nanoparticles, polarization parameters, alkane

## Abstract

The presented research shows the possibilities of creating in-line magnetic sensors based on the detection of changes of light propagation parameters, especially polarization, obtained by mixing Fe_3_O_4_ nanoparticles with hexadecane (higher alkane) surrounding a biconical optical fiber taper. The fiber optic taper allows to directly influence light parameters inside the taper without the necessity to lead the beam out of the structure. The mixture of hexadecane and Fe_3_O_4_ nanoparticles forms a special cladding surrounding a fiber taper which can be controlled by external factors such as the magnetic field. Described studies show changes of transmission (power, loss) and polarization properties like azimuth, and ellipticity, depending on the location of the mixture on sections of tapered optical fiber. The taper was made of a standard single-mode telecommunication fiber, stretched out to a length of 20.0 ± 0.5 mm and the diameter of the tapers is around 15.0 ± 0.3 μm, with the loss lower than 0.5 dB @ 1550 nm. Such a taper causes the beam to leak out of the waist structure and allows the addition of the external beam-controlling cladding material. The presented research can be used to build polarization switches or optical sensor. The results show that it can be a new way to control the propagation parameters of a light beam using tapered optical fiber and magnetic mixture.

## 1. Introduction

Nanomaterials have revolutionized the world of researchers. By combining nanoparticles (NPs) with various materials, it was possible to improve many material properties, which in turn increased the effectiveness of many devices, sensors, and improved parameters in various areas, including medicine [1,2], conversion energy [3,4], industry [5] and sensors [6], etc. Fiber optic technology also uses nanomaterials as a material doped to the fiber [7], or as a part of an extra layer on the fiber [8,9]. For different geometrical structures of optical fibers, scientists use many nanoparticles coatings and films [10,11]. The combination of these two technologies allowed the development of the non-telecommunications part—the construction of gas sensors [12] and the detection of other parameters such as pH [13], temperature [14,15], humidity [16], etc. They also allowed us to improve sensor parameters: switching time of liquid crystal cells [17], sensitivity [18], dynamic range [19], etc. By adding nanoparticles, it is also possible to enhance the non-linear and fluorescent effects [20]. One of the frequently used fiber optic structures for connecting with nanomaterials for non-telecommunication applications is tapered optical fiber technology (TOF) [21]. Thanks to TOF, we can expose the evanescing field to the outer sensitive coating that surrounds the TOF area. It is directly related to the change in the standard optical fiber geometrical parameters during the tapering technological process [9,22]. As the optical fiber is pulled out, the diameter of the fiber is reduced, and thus the part of the evanescent field emerging in different regions of TOF changes simultaneously. The most sensitive area in a taper is the waist—this is where the evanescent field is the largest, see Figure 1. In this way, the smallest changes in the refractive index (RI) parameter of the environment (sensitive coating) are detected by TOF (Figure 1).

The evanescent field is related to the penetration depth parameter (*d*_p_) (how deeply the light beam penetrates the cladding), which depends on the wavelength (λ), and the value of RI (*n*_core_, *n*_cladding_) for a standard fiber before the tapering process is described by the formula [23]:
$$d_{p}=\frac{\lambda }{2\pi \sqrt{n_{\mathrm{core}}{}^{2}\sin^{2}\theta_{i}-n_{\mathrm{cladding}}{}^{2}}}$$

In the taper waist area, the RI of TOF and the RI of sensitive coating should be taken into account to estimate the *d*_p_ value.

Materials engineering and fiber optic technology allow many combinations of different materials, which gives further interesting results and sensor application possibilities. The research presented in the article is an extension of the previously performed TOF tests and additional material. Research on this type of combination began with TOF and pure alkanes [24]. Due to the use of the properties of phase transitions of liquid–solid alkanes, which is directly related to the step change in the refractive index of the given alkane surrounding the taper, it was shown that the losses in the structure and ON–OFF transmissions were related to the change of temperature. This research showed the possibility of using this connection of materials as a temperature sensor. During the investigation, due to the low thermal conductivity of alkanes, a temperature hysteresis between the heating and cooling processes was observed. Hence, in a next stage of the research, ZnS: Mn nanoparticles of various concentrations were added to the alkanes [25,26]. Doping with NPs allowed for a change in temperature hysteresis and the possibility of a faster change from the ON–OFF state, and vice versa. NPs constituted additional centers of crystallization and additionally influenced the improvement of alkane parameters by increasing thermal conductivity. The research in this article focused on expanding the study and combining hexadecane (higher alkane C_16_) with magnetic Fe_3_O_4_ NPs nanoparticles. Thanks to such doping, we can control the position of the liquid with the help of a magnet, and, thus, influence the polarization parameters, which is related to the possible use of such a structure as an optical power sensor or a polarization parameter switcher. This article shows a new possibility to control parameters of light propagated in TOF, by changing the location of a new magnetic mixture (C_16_ + Fe_3_O_4_ NPs). The basic difference between the last and new research is changing the type of NPs, we used the magnetic properties of NPs which affect alkanes, without measuring thermal properties of the mixture as in [24,25,26]. In this article, we show the possibility of dynamically changing the polarization properties at room temperature. Parameter fluctuations are caused by a change in the location of the mixture between areas of the TOF, which causes a temporary change in the RI of the environment, whereas in the previous research, only a temperature change could influence transmission variations and polarization parameters. In the paper, the appropriate concentration of NPs in C16 was selected, a method of securing the TOF was developed, and technological parameters were selected for the production of TOF (such as elongation, consequently TOF diameter, losses) to obtain the best final results when TOF is connected with an additional material.

## 2. Materials and Methods

### 2.1. Materials

The basic material with which the Fe_3_O_4_ nanoparticles were mixed was hexadecane (C_16_H_34_) C_16_. It belongs to the group of higher alkanes, which are organic materials, described in detail in the literature [25,27,28,29]. The choice of such material was dictated by the most important parameter determining the environment surrounding the TOF which is the RI. This parameter determines the change of the boundary conditions which is directly related to changes in other parameters such as losses, polarization parameters, transmission, etc. The RI of pure alkanes varies abruptly depending on the phase transition temperature [27]. The liquid state of alkanes has a lower RI than the solid state. In general, for this group of organic materials, the temperature of the phase transition from solid to liquid increases with the carbon group extension. In our application, the important parameter is the RI below the RI of taper materials (1.46) because only then can light propagate in this structure. When the RI is higher, the light leaks beyond the taper area according to the total internal reflection principle [25]. During the research, we chose the C_16_ material which is liquid at room temperature. As a result, the position of the liquid can be changed without major obstacles and mechanical damage to the taper. The RI of a pure C_16_ alkane is lower than the RI of the taper (Table 1). In addition, a pure C_16_ at room temperature does not make a large loss over a wide range of wavelengths, unless the material becomes solid (temperature lower than 18 °C, according to Table 1); thus, the light will radiate from the taper and the power decreases to zero. Based on this information, it can be determined that the practical application of the combination of TOF with a mixture (C_16_ + Fe_3_O_4_ NPs) is possible when the temperature is higher than the transition temperature [24,25]. The changes in an ON–OFF mode for this connection can be directly related to the complex RI [30]:*n*’ = n + i ϰ,
where *n* determines the phase velocity and the coefficient ϰ (extinction coefficient) determines the losses. When the state of phase changes, the density changes, and the coefficient of excitation increases and causes greater losses.

The research idea was to surround a small part of a taper region which will influence a polarization property as a point disturbance of light propagation. The selected alkane was combined with magnetic NPs because without NPs there is no possibility of changing this material position and, thus, the parameters we want to influence would remain unchanged. The concentration of the mixture of an alkane and Fe_3_O_4_ NPs is 2 wt%. The choice of concentration was important because too small an amount would not allow for controlling the position of the entire C_16_ + NPs-Fe_3_O_4_ mixture, whereas too high a concentration could introduce large losses resulting from a possible agglomeration and deposition of NPs on TOF. Literature data reported, as well as our basic research, prove that the best results for combining materials with Fe_3_O_4_ NPs are for concentrations in the range of 1–2 wt.% [32]. The preparation of Fe_3_O_4_ NPs started from a synthesis of iron trioleate [33,34]. A second step was a synthesis of Fe_3_O_4_ NPs upon heating to 320 °C for 30 min, precipitated by adding ethanol and centrifuged in the mixture of ethanol/toluene (1:1, *v/v*). The nanoparticles were dried and redispersed in chloroform and measured with Dynamic Light Scattering (Malvern ZS Nanosizer, Malvern, UK), measurements. The size distributions are shown in Figure 2.

To procees full characterization of Fe_3_O_4_NPs we analized Powder X-ray diffraction (Figure 3) as well as make the picture on Scanning Transmission Electron Microscopy (STEM) (Nova NanoSEM 450, Lincoln, NE, USA).

The diffractogram reveals the structure typical for Fe_3_O_4_ [34,35]. The size of the crystals calculated according to the Scherrer equation [36] is 7.00 ± 1.42 nm. The difference between PXRD size and DLS measurements was observed because DLS measures the hydrodynamic radius of the nanoparticles in the suspension, whereas PXRD receives information about the size of single crystallites. For polydispersed samples, the broadening of the peaks in PXRD measurements highly depends on the fraction of the smaller crystallites, whereas in DLS, the measurement may be biased by the presence of the fraction of bigger objects, which scatter light intensively.

The picture form Scanning Transmission Electron Microscopy of used magnetin naoarticles Fe_3_O_4_ is presented in Figure 4.

The measurement of Zeta Potential is presented in Figure 5.

The suspension was stable due to the interaction between ligands and solvent molecules. The STEM pictures were taken with Nova NanoSEM 450 (Lincoln, NE, USA). PXRD measurements were performed using Panalytical Empyrean. Zeta potential and DLS were measured on Malvern Zetasizer, Malvern, UK.

Nanoparticles prepared according to the mentioned methods were mixed with alkanes in the following way: nanoparticles were weighed on an analytical weight (20 mg) and transferred to a glass vial with a capacity of 1.5 mL. Nanoparticles were poured with 1.0 mL of hexadecane and placed in an ultrasonic washer for 30 min at room temperature. After combining, the mixture of C_16_ + Fe_3_O_4_ NPs was characterized by RI—1.4366. RI was measured with a HANNA—HI 96800 electronic refractometer for the wavelength: sodium line D: 589 nm, at room temperature.

### 2.2. Technology

To create an in-line optical fiber sensor, an evanescence wave should be obtained to interact with a magnetic mixture. For this purpose, the optic fiber taper technology was applied. Tapers were made using a fiber optic taper element technology (FOTET) set-up, which is widely described in different papers in [24,25,26,37]. Manufactured TOF on FOTET is described by parameters such as: an elongation, a diameter of TOF, a torch move length and a loss for 1550 nm. The optimized parameter of TOF allows the connection with an extra material and gives a possibility to influence parameters of the propagated light with a low loss. For different coating materials, the parameters of TOF should be properly selected because the extra materials are characterized by different optical properties and use various physical phenomena. Movement of torch gives a possibility to manufacture TOF with different TOF regions lengths [37] (Figure 6). When the torch is without the movement, then the taper waist—length and diameters—is smaller than when torch is moved by a few millimeters. The movement of torch ensures a longer interaction time with the external environment, which is necessary for the application of additional material.

The optimization for these sensors was connected with choosing the appropriate length and diameter of a taper which makes it possible to interact and observe the influence of external material on light propagation. The tapers characterized by an elongation in the range of 15–30 mm were tested and for an elongation of about 20 mm, the best results were obtained. The most important parameters of the taper used in measurements are: pulled out to a length of 20.0 ± 0.5 mm, which corresponds to the diameter of the taper being around 15.0 ± 0.3μm, measured losses are lower than 0.5 dB @ 1550 nm, move of torch 5 mm. The mentioned taper with a given diameter is described in the literature as a double-clad [24,25,26], which means that the created structure retains both the core and the cladding in its structure, and the alkane with NPs forms the second cladding. Below the mentioned taper diameter, the core does not play the leading role, the whole taper waist becomes the leading role, and the surrounding environment becomes a single cladding. According to the principle of propagation, only for selected RIs of the cladding propagation is possible, for the others large losses occur. The implementation of a taper with a larger diameter makes it unable to have direct access to the evanescence wave and prevents changing the wave parameters.

An important technological element for testing is the properly built protection of the taper structure, which protects the TOF from breaking, and contamination, also enables a practical connection with the selected material, and, most importantly, allows the connection to an external magnetic field (Figure 7). For this purpose, a glass tube with holes was used so that the mixture could be poured in, and then the holes were sealed with a NOA76 UV light-curable adhesive. Additionally, the transparent tube made it impossible to determine the position of the mixture when changing the magnet. The tube was refilled with 0.03 mL C_16_ + Fe_3_O_4_ NPs using a laboratory pipette. The measurement was carried out at room temperature. Two N42 magnets with dimensions of 10 × 10 × 10 mm were used with the following parameters: remanence induction Br: 1.28–1.32 [T], coercivity HcB: min. 923 [kA/m], coercivity HcJ: min. 955 [kA/m], magnetic energy density (BH) max: 318–342 [kJ/m^3^].

The transmission measurement system used consisted of: a broadband light source NKT Photonics, SuperK Extreme EXR-15, Birkerod, Denmark (operating in the 400–2400 nm range), a splitter connected to the NKT PhotonicsSuperK SPLIT laser range NIR/IR and VIS/NIR, optical spectral YOKOGAWA analyzer model AQ6373, Tokyo, Japan (operating in the range of 350–1200 nm) and AQ6375, Tokyo, Japan (operating in the range of 1200–2400 nm) (Figure 8).

Polarization measurements were carried out by using two configurations of the set-up presented in Figure 9. The first configuration consists of the light source (TSL-210V, Santec, Aichi, Japan) deterministic polarization controller (DPC 5500, Thorlabs, Newton, NJ, USA), sample, and polarimeter as two parts, i.e., head (PAN5710IR3, Thorlabs, Newton, NJ, USA) and electronic unit. Both DPC and polarimeter electronic units were integrated into the same chassis (TXP 5005, Thorlabs, Newton, NJ, USA). This set-up allows to measure with a polarimeter changes generated by the DPC Stokes vectors. Acquired data were used to calculate Mueller matrices of the tested sample. Using the Lu-Chipman decomposition method [38], it is possible to extract from the Mueller matrix all optical parameters of the sample, i.e., losses, depolarization, dichroism, and birefringence [37,38]. In the simplified version of this set-up, when DPC was excluded from the set-up, only the Stokes vector changes were measured. In this measurement, two different wavelengths of light sources, i.e., 850 nm (LDPC-850, Optics, Ottawa, Canada), and 1550 nm (TSL-210 Santec, Aichi, Japan) with appropriate polarimeter heads, i.e., PAN5710IR2, and PAN5710IR3 (Thorlabs, Newton, NJ, USA) were applied. Data were acquired and processed by PC.

## 3. Results

Transmission measurements are presented in Figure 10 for the two wavelength ranges of 500–1200 nm and 1200–2000 nm.

As can be seen, the applied material works in a wide spectral range. The observed transmission losses result from the deposition of NPs on TOF and higher RI of the mixture than RI of air. For pure alkanes at room temperature, such effects are not noticed [24]. As the mixture is moved, an active drop in power is visible—the constriction responds to the change in rolling RI. Locating the mixture in the TOF region causes losses for the entire wavelength range.

A broader analysis of the optical properties of TOF with magnetic NPs mixture was carried out based on polarimetric studies. Using the first configuration of the set-up presented in Figure 6 and applying the measurement sequence shown schematically in Figure 8, changes of the generated specific states of polarization were measured and Mueller matrices were further calculated at each of indicated positions of the magnet [39,40]. In Figure 11, numbers from 0 to 4 indicate the positions at which the magnet was held, and the measurement was performed. The magnet was sequentially moved from the left to right and backward. For positions 0, 1, 3, and 4, the mixture was far from TOF, whereas for position 2, the nanomaterial covers it.

Calculated Mueller matrices consist of full information about the optical properties of the tested sample. By using the standard Lu-Chipman decomposition method, five main optical parameters, i.e., losses, depolarization, dichroism, optical rotation, and linear retardance were calculated [39,40]. Depolarization changes were not observed for this optical element, however, four remained parameters experienced significant changes. These parameters were shown in Figure 12. Vertical axes of all plots in these plots indicate magnet position correlated with Figure 11. The points of these plots are measured points, whereas the solid lines are interpolations between these points.

Changes in all parameters presented in Figure 12 show that when the light passes through the optical fiber far from TOF (positions 0, 1, 3 and 4) only optical rotation is disturbed after the first exposition of TOF on the mixture. Losses, dichroism, and linear retardance are kept at almost the same values in these positions. However, when the light passes through TOF with an additional cladding created by the mixture, all calculated parameters change significantly. The dynamic range of losses is about 15 dB and the dichroism changes were at 0.38, whereas optical rotation and linear retardance were about 0.35 rad and 0.32 rad. These changes are significant and can be easily measured. Moreover, it is easy to distinguish when the TOF is covered with the mixture for the first time, i.e., the first change of each of the presented parameters on the left part of each plot. In all parameters, their change when the TOF is cladded the second time is reduced. This is due to some mixture remaining as a thin film on the taper waist.

The polarimetric measurements presented above are static ones. Therefore, in further studies, measurements were carried out without using the DPC subsystem and time changes of the Stokes were acquired (dynamic measurements). The results were analyzed with the use of optical power, azimuth, and ellipticity.

During the dynamic measurements, the time when the mixture is in the taper waist area was changed, causing changes in the polarization parameters. The magnets were moved along the taper at a constant speed, setting the mixture on the taper waist for a period of 5 s, 10 s, or rapid sweep causes a temporary change. Transmission changes and losses may be caused by the fact that some NPs may be deposited on the taper during the liquid sweep and temporary changes of RI.

The general principle of operation in a dynamic measurement is presented in Figure 13. The time of interaction with the mixture influences changes in parameters, and then when the mixture is outside the taper waist area, the parameters return to their original state. Measurements were carried out using the second configuration of the set-up presented in Figure 9. Figure 14 shows the reference values of the polarization parameters (optical power, ellipticity, and azimuth) before applying the mixture.

The first measurement was linked with measuring the optical power of light propagating in a taper. In Figure 15, the power change depends on the appearance and disappearance of a higher alkane with magnetic nanoparticles around the chosen region of an optical fiber taper by applying a magnetic field. Measurements were provided for two wavelengths of 850 nm and 1550 nm.

As it can be noticed for both wavelengths, the movement of higher alkanes doped with magnetic NPs causes a decrease in power. For all measurement cases, fast movement of the mixture, 5 s and 10 s position of the mixture on a taper waist region are observed which results in a power decrease. For a wavelength of 850 nm, the dynamics of changes are from 0.2 dBm to1.0 dBm. Additionally, for the shorter effect of the magnetic mixture effect, the power shapes are irregular. Such small changes are linked with a low impact of evanescence wave with an external magnetic mixture which confirms the theory of deep penetration of a leaking wave for this wave and single-mode fiber for the telecommunication range. For a wavelength of 1550 nm for all cases, the dynamics of change are much higher than for 850 nm and oscillate around 15 dBm and the shapes are stabilized for an extended period. These results prove the proper match of taper parameters which allowed the magnetic mixture to interact with the evanescence wave. In all graphs, a significant disturbance of power (peaks) can be observed. There are related to a displacement of the magnetic mixture through the transition area, as well as a sudden change of RIs for the taper waist area. This effect will be investigated in future measurements.

For this measurement setup, the change of azimuth and ellipticity was investigated. Figure 16, Figure 17 and Figure 18, show the graphs describing the changes in the above-mentioned parameters (optical power, azimuth, and ellipticity) for a rapid change in the RI of the external environment of TOF caused shifting of a mixture under the influence of the magnetic field for a different time—5 s and 10 s for wavelengths of 850 nm and 1550 nm, respectively.

Analysing the results from the figure above, it can be seen that when we change the position of the magnetic mixture on the taper region, from untampered to taper waist, the polarization parameters like azimuth and ellipticity change rapidly. As it was mentioned above, for 1550 nm, a higher dynamic of change for both parameters (over 20 times) can be obtained in all cases than for an 850 nm wavelength. For all cases, disturbance on slopes connected with matching RIs and transitions through the transition region can be seen. The time of changing the parameters is strictly connected with the movement of the mixture along the taper. For the length of 1550 nm, significant disturbances and lack of stability of the measured parameters (discontinuous curves) can be seen, which additionally proves the interaction of the nanoparticles themselves with the wave and the interaction within them. During the measurements, it was also noticed that a part of the material after a single pass may remain in the area of the taper, creating a very thin layer of the magnetic mixture which causes additional disturbances, as a result of which it may distort the effect for the next adjustment of the mixture to the proper area. In future measurements, detailed studies on the time of moving the mixture to the proper area will be carried out, and the analysis will include the fluctuations occurring on the slopes of the runs. The results of the presented tests are reproducible.

## 4. Conclusions

The presented results show the possibility to manufacture in-line optical fiber devices/sensors of optical power or polarization parameters based on an optical fiber taper surrounded by a magnetic mixture. The main conclusions of the experimental studies can be as follows:-It is possible to design sensors/devices working in a wide range of 500–2000 nm at room temperature for different wavelengths;-A single-mode fiber should be applied with the cut-off wavelength correlated with this wavelength to prevent multi-mode operation and increase the dynamic range;-It is possible to manufacture in-line optical fiber sensors of the magnetic field with a high dynamic range of losses (of) about 15 dBm @ 1550 nm;-The mixture used has well-chosen optical parameters; it does not make large losses to the structure at room temperature;-By using the C_16_ + NPs Fe_3_O_4_ mixture and controlling its position in different TOF regions, there is the possibility to influence or change polarization parameters, i.e., ellipticity or azimuth;-Transmission changes and losses may be caused by the fact that some NPs may be deposited on the taper during the liquid shift;-The best and the biggest changes for a 1550 nm wavelength were obtained. For shorter waves, the changes were smaller which is connected to a smaller penetration depth, and thus the light is more trapped in the TOF structure; thus, there is less influence of external factors on the propagating light beam.

The conducted research was aimed at presenting the possibility to design and creating a new kind of sensor detecting a magnetic field that can be observed by changing the optical parameters with a high dynamic range. In this article, the changes of parameters of propagated light are strictly connected with the location of a magnetic mixture moved by an applied magnetic field—in contrast to the earlier studies where the changes were caused by temperature. Additionally, the change in polarization parameters was observed depending on the position of the magnetic mixture, showing the possibility of creating a low-cost polarization-sensitive sensor depending on the magnetic field.

The measurement was carried out at room temperature where C_16_ is in a liquid phase. The solid phase does not allow for the shift of a mixture for a chosen region of TOF. In future research, we will provide additional measurements of the influence of the temperature shift in which we will test how the NPs and alkanes will behave. The tests will check the temperature range in which we retain the NPs shifting effect while maintaining the influence on polarization and transmission properties without major losses. In the next stage of research, we will change the type of fiber from SMF to polarization-maintaining fibers and use various magnetic mixtures, including mixtures of liquid crystals and magnetic NPs.

## Figures and Tables

**Figure 1 sensors-22-07801-f001:**
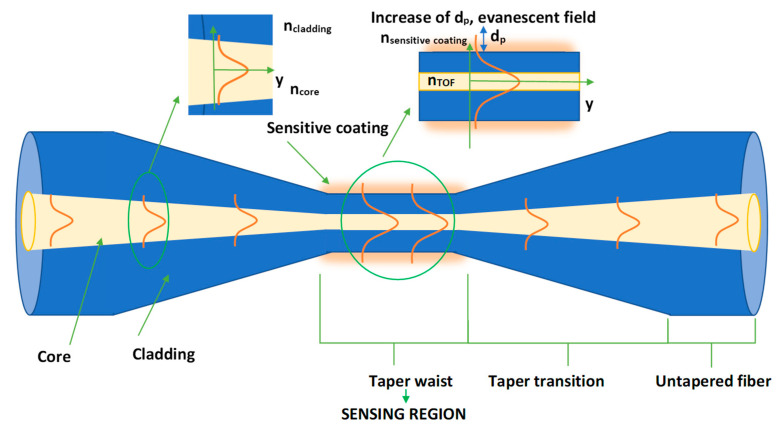
Scheme of the tapered optical fiber with the main region marked and visualization of the change of the propagating wave inside with evanescent field in this region.

**Figure 2 sensors-22-07801-f002:**
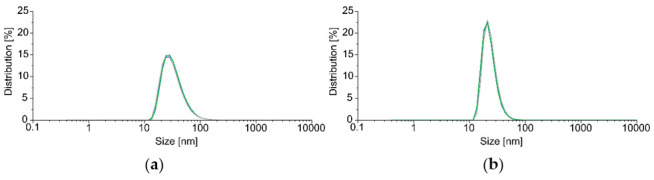
Size (diameter) distribution of Fe_3_O_4_NPs redispersed in chloroform (20 °C) (**a**) by volume (**b**) by number. Measurements were conducted in three repetitions (visible as green, blue, and gray lines). The average sizes for the average measurement were 34.32 nm with a standard deviation of 17.79 nm by volume and 23.49 nm with a standard deviation of 7.66 nm by number.

**Figure 3 sensors-22-07801-f003:**
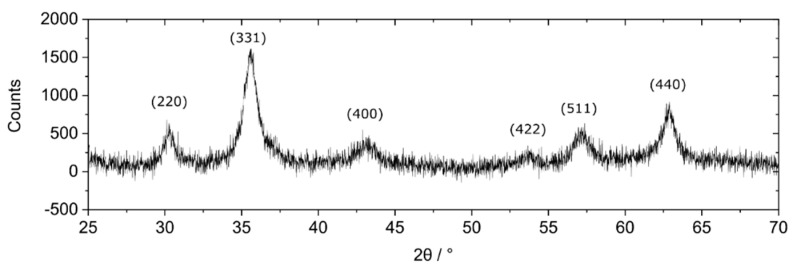
Diffractogram of magnetic nanoparticles. The background was subtracted from the presented data Cu K(α).

**Figure 4 sensors-22-07801-f004:**
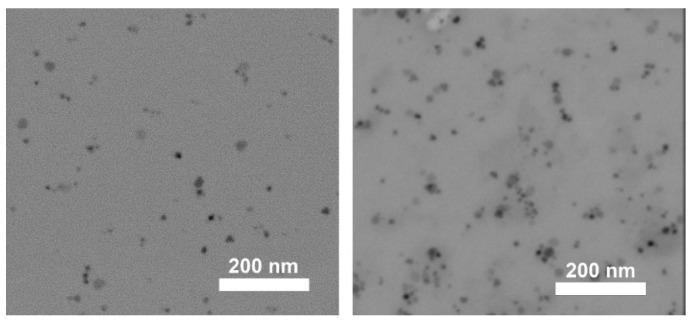
STEM (scanning transmission electron microscopy) pictures of Fe_3_O_4_ nanoparticles deposited on a holey carbon film.

**Figure 5 sensors-22-07801-f005:**
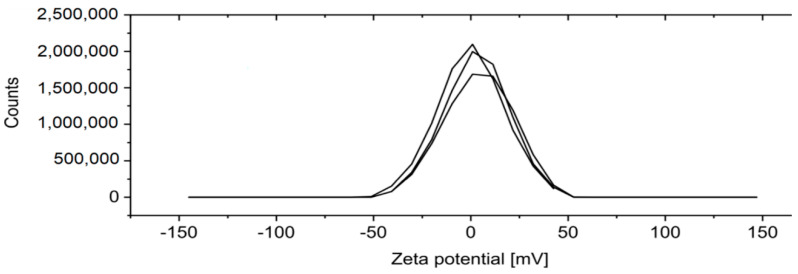
Zeta potential distribution of Fe_3_O_4_ nanoparticles. Three repetitions were conducted in toluene, 25 °C.

**Figure 6 sensors-22-07801-f006:**
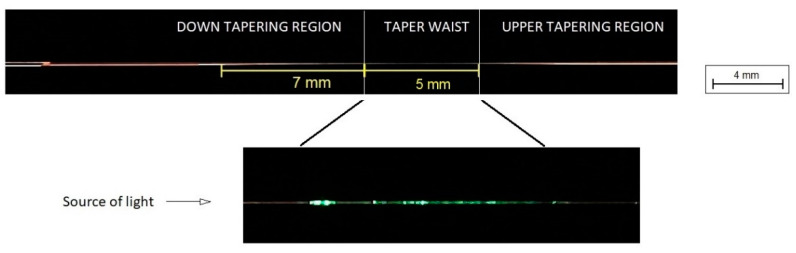
Tapered Optical Fiber—manufactured on FOTET.

**Figure 7 sensors-22-07801-f007:**
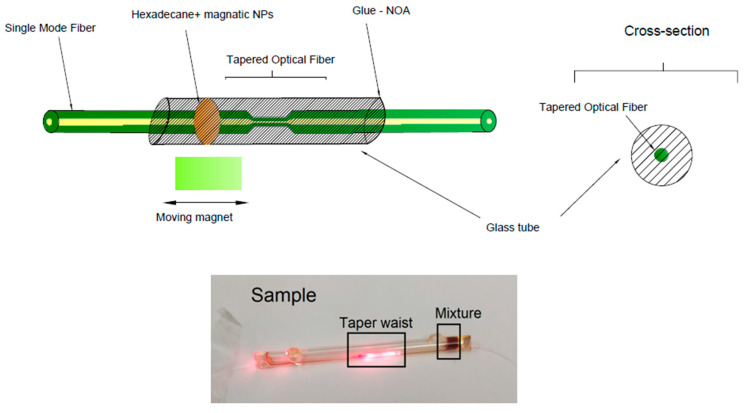
Scheme and picture of the prepared magnetic sensor based on a taper surrounded with C_16_ + Fe_3_O_4_ NPs mixture.

**Figure 8 sensors-22-07801-f008:**
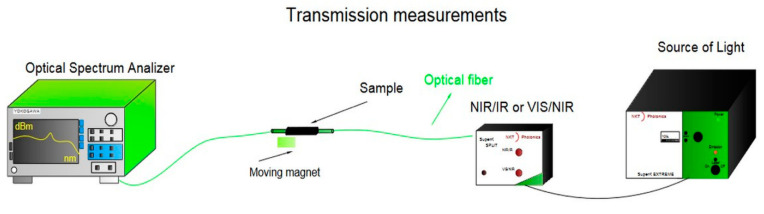
The arrangement for measuring the transmission of a magnetic sensor in the broadband range of 500–2000 nm.

**Figure 9 sensors-22-07801-f009:**
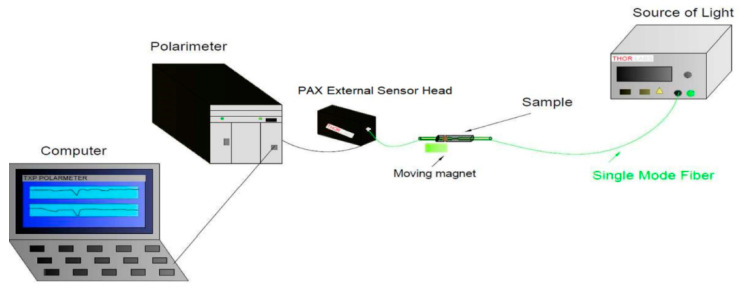
The arrangement for measuring the polarization parameters of a magnetic sensor for different wavelengths of 850 nm and 1550 nm.

**Figure 10 sensors-22-07801-f010:**
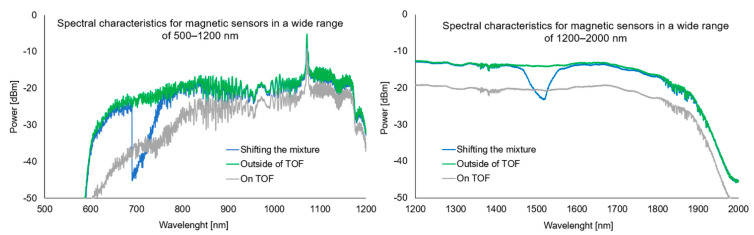
Spectral characteristics for magnetic sensors in a wide range of 500–1200 nm and 200–1200 nm with a mixture outside the taper region (green line), on the taper region (grey line), and visualization of the sliding mixture along the taper (blue line).

**Figure 11 sensors-22-07801-f011:**
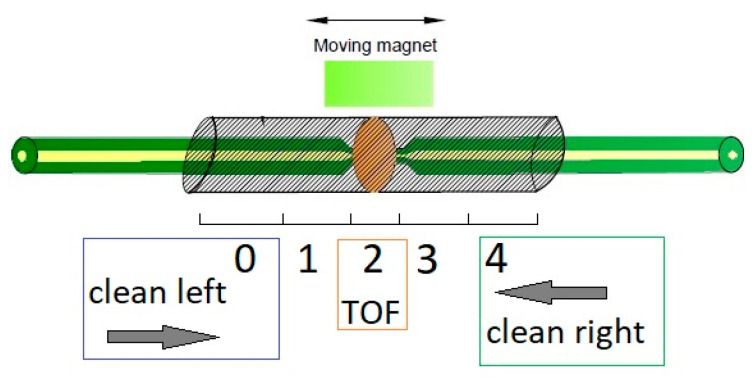
Measurement scheme for determining the effect of the magnetic mixture on the TOF properties. Measurements were started from position “0” and the magnet was moved to position “4” from the left of TOF to the right and backward with the measure in the same position. At each of these points, the Mueller matrix was calculated.

**Figure 12 sensors-22-07801-f012:**
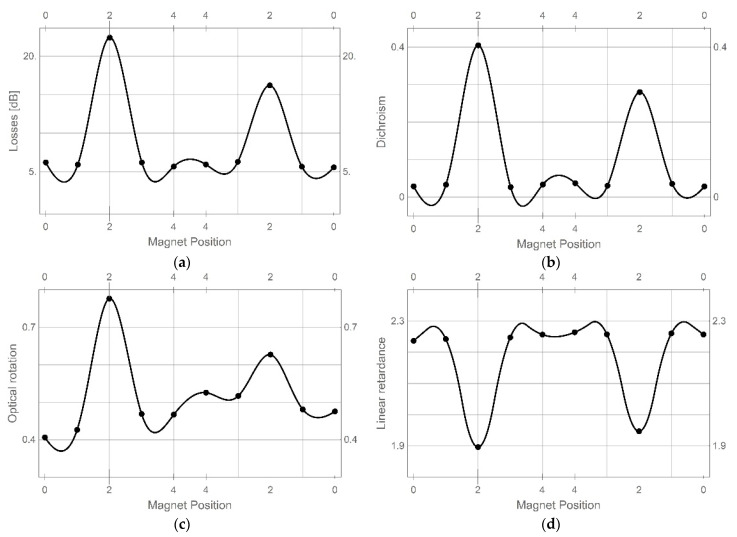
Optical parameters of the tested sensor: (**a**) losses, (**b**) dichroism, (**c**) optical rotation, and (**d**) linear retardance calculated for positions indicated in Figure 7.

**Figure 13 sensors-22-07801-f013:**
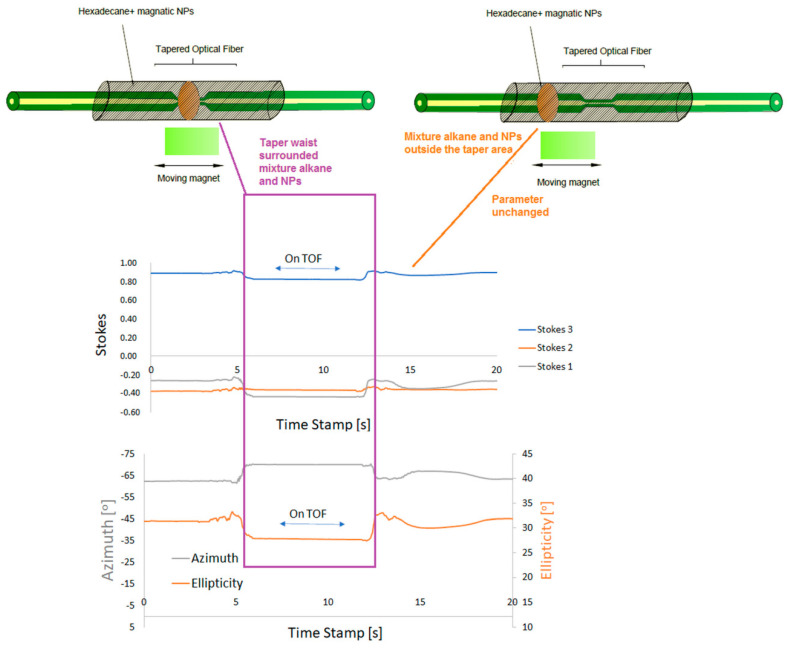
Scheme, idea, and results of the proposed measurement of the magnetic sensor.

**Figure 14 sensors-22-07801-f014:**
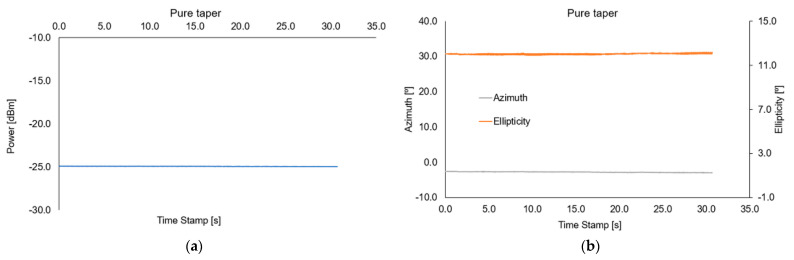
Reference graphs of (**a**) optical power, (**b**) azimuth and ellipticity parameters for the tapered optical fiber taper.

**Figure 15 sensors-22-07801-f015:**
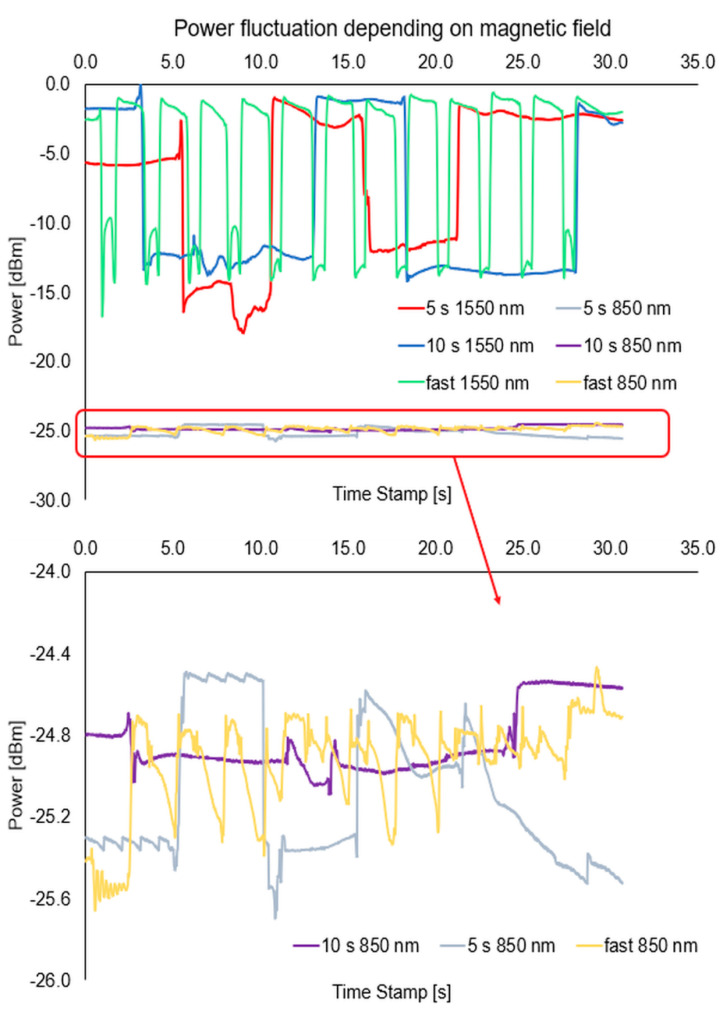
Power changes depending on the appearance and disappearance of a higher alkane with magnetic nanoparticles around the chosen region of an optical fiber taper by applying a magnetic field for 850 nm and 1550 nm wavelengths.

**Figure 16 sensors-22-07801-f016:**
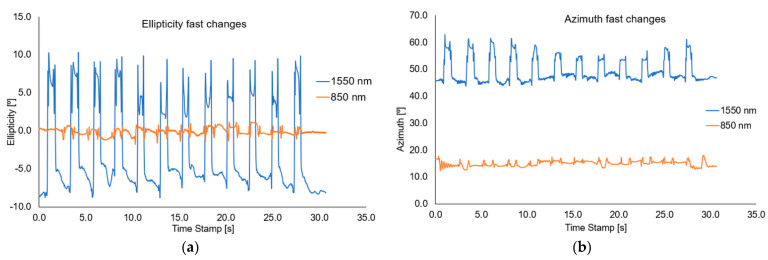
Graphs of influence of sliding the magnetic mixture on change the ellipticity (**a**) and azimuth (**b**) for a rapid sweep of magnetic field for wavelengths of 850 nm and 1550 nm.

**Figure 17 sensors-22-07801-f017:**
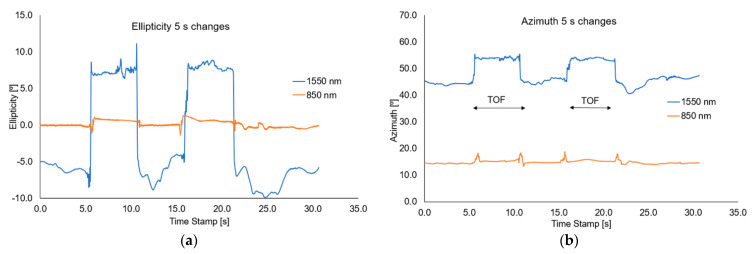
Graphs of influence of shifting the magnetic mixture on the change of ellipticity (**a**) and azimuth (**b**) for 5s reaction of magnetic field for wavelengths of 850 nm and 1550 nm.

**Figure 18 sensors-22-07801-f018:**
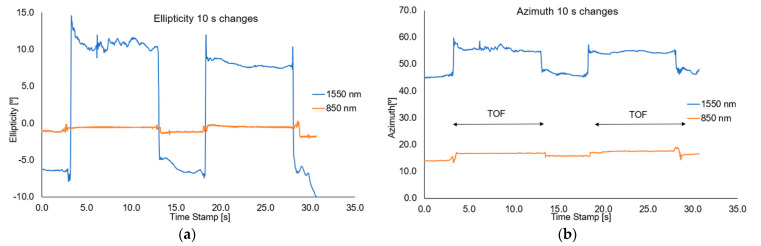
Graphs of influence of shifting the magnetic mixture on the change of ellipticity (**a**) and azimuth (**b**) for 10 s reaction of magnetic field for wavelengths of 850 nm and 1550 nm.

**Table 1 sensors-22-07801-t001:** The main parameters of hexadecane materials [25,31].

Material	RI (589 nm, 20 °C)	Phase Transition Temperature (°C)
C_16_ *n*-Hexadekan (C_16_H_34_)	1.4345	18

## Data Availability

Not applicable.

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
