# Peer review of "Optical Properties of a Tapered Optical Fiber Coated with Alkanes Doped with Fe3O4 Nanoparticles"

_sensors, 2022, doi:10.3390/s22207801_

Round 1
Reviewer 1 Report
The article reported by Karol et al. entitled “Research on optical properties in a tapered optical fiber with higher alkanes cladding doped with Fe3O4 nanoparticles” report an experimental demonstration of a magnetic sensor utilizing tapered fiber configuration. I've been through some similar research articles reported by the same group and found major flaws. Therefore, I don't think it is acceptable in its present form. Comments are given below:
1. The sensing response is not comparable to the author's own work using ZnO and LC published in recent years. (10.3390/mi11111006,10.3390/cryst9060306)
2. I didn't find optimization of the structural parameters in any of the previous articles. What is the optimization rule for device fabrication? size of nanoparticles, tapering length, and down and upper tapering region.
3. I didn't see any material characterization and SEM data regarding the size of the nanoparticles, tapering length, and upper and down tapered region for its validation.
4. How many times author performed the experiment? what was the maximum deviation observed during each trial?
Author Response
We would like to thank the Reviewer for their time and effort in carefully checking our manuscript. Being very grateful for all valuable comments, insights, and advice, we have made the suggested changes in the manuscript and prepared answers for all comments. The answers for all comments are attached in a pdf file.

Reviewer 2 Report
It seems that the paper missed something. Authors didnot illustrate it in the introduction. However i have some issue as follows.
1. Authors used Fe3O4 NPs in this paper. It lacks characterization of NPs.
2. Authors seems taking advantage of a supercontinuum spectrum source, do they consider the thermal effect during the test?
3. How did the strength of the applied magnetic field affect the performance?
Author Response

(The authors gave the same response as above.)

Reviewer 3 Report
The authors should improve the introduction as well as the references
The innovations have to be highlighted in the conclusion and abstract
English need revision
Further investigations should be added in order to show the possible continuation of the research
Author Response

(The authors gave the same response as above.)

Round 2
Reviewer 1 Report
The author has addressed all the raised concerns positively, hence I recommend its possible publication in Sensors. Good luck
Reviewer 2 Report
Authors give a postive response.